# Innovative Phosphorene Nanoplatform for Light Antimicrobial Therapy

**DOI:** 10.3390/pharmaceutics15122748

**Published:** 2023-12-09

**Authors:** Elisa Passaglia, Antonella Sgarbossa

**Affiliations:** 1National Research Council-Institute of Chemistry of OrganoMetallic Compounds (CNR-ICCOM), SS Pisa, Via Moruzzi 1, 56124 Pisa, Italy; elisa.passaglia@pi.iccom.cnr.it; 2National Research Council-Nanoscience Institute (CNR-NANO) and NEST-Scuola Normale Superiore, Piazza S. Silvestro 12, 56127 Pisa, Italy

**Keywords:** antimicrobial, phosphorene, photothermal therapy, photodynamic therapy

## Abstract

Over the past few years, antibiotic resistance has reached global dimensions as a major threat to public health. Consequently, there is a pressing need to find effective alternative therapies and therapeutic agents to combat drug-resistant pathogens. Photodynamic therapy (PDT), largely employed as a clinical treatment for several malignant pathologies, has also gained importance as a promising antimicrobial approach. Antimicrobial PDT (aPDT) relies on the application of a photosensitizer able to produce singlet oxygen (^1^O_2_) or other cytotoxic reactive oxygen species (ROS) upon exposure to appropriate light, which leads to cell death after the induced photodamage. Among different types of 2D nanomaterials with antimicrobial properties, phosphorene, the exfoliated form of black phosphorus (bP), has the unique property intrinsic photoactivity exploitable for photothermal therapy (PTT) as well as for PDT against pathogenic bacteria.

## 1. Introduction

In 2023, the World Health Organization (WHO), in partnership with the Global AMR R&D Hub, released a report for G7 Finance and Health Ministers detailing progress on incentivizing the development of new antibacterial treatments [1]. The WHO 5-year strategic plan—the 13th General Programme of Work 2019–2023 (GPW 13) included the emergence of antimicrobial resistance among the ten main threats to global health, with the deaths of 4.95 million people in 2019. The discovery and the development of antimicrobial drugs such as antibiotics probably represents one of the greatest successes of modern medicine. However, the rapid adaptability of microorganisms, caused mainly by genetic mutations and a fast reproduction rate, makes these drugs almost ineffective in fighting infections.

Furthermore, the overuse and misuse of antibiotics have accelerated the emergence and spread of antimicrobial resistance, which will account for an estimated 10 million deaths by the end of 2050 [2]. The “golden era” of antibiotics is running out, as increasing reports of multidrug-resistant pathogens send us back to a time when we were unable to easily treat common infections. New resistance mechanisms are being described constantly, and new genes and vectors of transmission are discovered regularly [3]. The reduced permeability of antibiotics, the increased efflux pumps, the modification and protection of antibiotic targets, and the inactivation of antibiotics by hydrolysis or by the transfer of a chemical group are the main molecular mechanisms associated with bacterial resistance [4]. In addition, the organization of microorganisms in biofilm (aggregates of microorganisms in which cells are frequently embedded in a self-produced matrix of extracellular polymeric substances) provides an increase in the tolerance of microorganisms against conventional antimicrobial agents [5].

Regarding comprehensive and sustainable actions to tackle antimicrobial resistance and related specific pathogens, it is crucial to support and promote the research and development of new antimicrobial therapies capable of efficiently inactivating pathogens without the risk of inducing resistance. As a consequence, many research efforts have been devoted to finding a novel non-antibiotic treatment. Antimicrobial photodynamic therapy (aPDT) has been suggested as a promising approach to treat microbial infections due to its spatiotemporal selectivity, non-invasiveness, minimal side effects, and broad antimicrobial spectrum. APDT utilizes non-harmful molecules, i.e., photosensitizers (PS), which are able to generate cytotoxic reactive oxygen species (ROS) upon irradiation at specific wavelengths of harmless visible light [6]. Numerous molecules have been suggested and evaluated as photoactive antimicrobials, with many of them coming from nature, for example, curcumin, hypericin, and flavin derivatives. In addition, tetrapyrrole compounds such as porphyrins, chlorins, bacteriochlorins, and phthalocyanines or synthetic dyes such as phenothiazinium, squaraine, and BODIPY (boron-dipyrromethene) and transition metal complexes and nanomaterials have been demonstrated to possess significant photo-killing effects against several bacterial pathogens [7,8].

Low-dimensional materials (LDMs), ranging from a few to one hundred nanometers and including carbon, graphene, graphene oxide (GO), black phosphorus (bP), boron nitride, molybdenum disulfide (MoS_2_), zinc oxide (ZnO), titanium, and copper oxide, possess unique physicochemical properties that make them excellent antimicrobial agents either alone or in combination with other materials.

The antimicrobial mechanisms of LDMs are thought to arise from their size, shape, chemical functionality, and surface properties that frequently provide a synergistic effect [9,10]. Among emerging 2D nanomaterials for potential biomedical applications, phosphorene has a peculiar feature: It possesses an intrinsic photoactivity exploitable for photothermal and photodynamic antibacterial treatment, i.e., aPTT and aPDT, respectively [11].

Phosphorene is a 2D nanomaterial obtained by exfoliation of bP; however, for the sake of clearness, the name phosphorene, which is commonly accepted and used by the scientific community, has no relation to the chemical nature of the material but is instead purely based on the conceptual similarity to graphene. As clearly stated, phosphorene, unlike graphene, is sp^3^-hybridized, and according to the IUPAC nomenclature, it belongs to the phosphane group; it is also correctly named 2D-phosphane [12]. Furthermore, as discussed later, obtaining individual flakes of a perfect monolayer bP still poses a synthetic challenge. Therefore, researchers usually study and apply nanoparticles that have from 2 up to 8–10 layers to their fields of interest, including biomedicine. In this review, we refer to a monolayer or a few layers of bP as “bP-NPs” (black phosphorus nanoparticles) and to the native form of bP as “bulk black phosphorus”.

Few-layered bP nanosheets have been shown to be effective photosensitizers exhibiting a broad-spectrum light absorption ranging from the UV light to the near-infrared region [13], producing singlet oxygen (^1^O_2_) with a high quantum yield [14]. In addition to these favorable properties, bP-NPs are biocompatible and nontoxic materials, which is the essential criteria for in vivo applications [15,16].

## 2. Antimicrobial Phototherapy

In antimicrobial phototherapy (APT), the light is used to treat infections caused by bacteria, viruses, fungi, and other microorganisms. This treatment method harnesses the properties of light, either alone or in combination with suitable photosensitizing agents, to induce a phototoxic or photothermal effect on the targeted microbes. APDT, also called photodynamic inactivation (PDI), and aPTT are antimicrobial approaches that have been shown to effectively inactivate a wide range of pathogens, including microorganisms that are highly resistant to conventional drugs and that form biofilms. Both therapeutic methods exploit the interaction of a photosensitizer with light to produce, in the case of aPDT, either highly reactive oxygen species in the presence of oxygen or, in the case of aPTT, heat and local temperature rise. The synergistic use of aPDT and aPTT can enhance the overall efficacy of antimicrobial phototherapy, leading to significant reductions in bacterial pathogens, eradication of infections, and collapse of biofilms.

### 2.1. Basic Mechanisms

Since many exhaustive reviews have recently described the latest updates on aPDT or PDI [6,7,8,17,18,19,20,21], herein, we will only recall some basic concepts that are useful for understand how aPDT works. The fundamental mechanism of aPDT is based on the simultaneous presence of three main actors: (1) a photosensitizer (PS), a molecule or a complex that is physiologically harmless and able to absorb light in spectral regions ranging from the visible to near-infrared wavelengths; (2) a light source emitted in the spectral range absorbed by PS; and (3) molecular oxygen, i.e., O_2_ [22] (Figure 1).

PS in its ground singlet state has two electrons with opposite spins. Following the absorption of a photon of light with the appropriate quantum energy (wavelength), one electron moves to a higher-energy orbital. The PS singlet excited state is very unstable and decays back to the ground state, losing energy either as the emission of light (fluorescence) or production of heat (internal conversion). It may also undergo a process known as “intersystem crossing” to form a more stable excited triplet state with parallel spins. The triplet-state PS molecule can decay back to the ground state by emitting light (phosphorescence) even though this is a “forbidden process” according to quantum selection rules. Therefore, the triplet state, having a lifetime of microseconds, is much more stable than the singlet state, which only lasts nanoseconds. The longer lifetime of the PS triplet state allows it to collide and react with molecular oxygen in its triplet ground state, transferring its energy. This energy transfer leads to the formation of singlet oxygen (^1^O_2_) and ground-state PS, giving rise to a type II photochemical process. Alternatively, a type I reaction occurs when PS in its excited triplet state undergoes an electron transfer (acquisition or donation of an electron) to form PS radical ions that, in turn, react with oxygen to eventually form reactive oxygen species (ROS) such as superoxide, hydrogen peroxide, and/or hydroxyl radicals (Figure 1).

In the case of PTT, the absorbed NIR light (700–1300 nm) is released as heat through different mechanisms of photothermal conversion depending on the photothermal agent used. These mechanisms are surface plasmon resonance in metals, electron–hole generation and relaxation in semiconductors, and HOMO (highest occupied molecular orbital)-LUMO (lowest unoccupied molecular orbital) excitation and lattice vibration of molecules [23].

Damage to bacterial cells or other microorganisms can be caused by heat and/or ^1^O_2_ and other ROS through various mechanisms, such as the oxidation of membrane lipids and amino acids in proteins, cross-linking of proteins, and oxidative damage to nucleic acids, resulting in pathogen inactivation.

### 2.2. Antimicrobial PDT Activity

To obtain an effective inactivation of more than 5 log_10_ steps of CFU (colony-forming units), which is considered a disinfecting action according to infection control guidelines, it is necessary to fulfill certain requirements. Due to the ^1^O_2_ limited diffusion during its lifetime, exogenous PS should be located very close to the bacterial target to exert a cytotoxic effect. As microbial cells have a negative charge that is more pronounced compared to mammalian cells, cationic PS molecules have been reported to be rapidly uptaken and bind selectively to bacterial cell walls. It is worth mentioning that aPDT has been shown to be particularly effective when the target microorganism naturally produces and accumulates endogenous photosensitizers, mainly porphyrins and flavins, such as in the case of *Candida albicans* [24], *Pseudomonas aeruginosa* [25,26], *Helicobacter pylori* [27,28,29], *Legionella rubrilucens* [30], *Neisseria gonorrhoeae* [31], *Porphyromonas gingivalis* [32,33], methicillin-resistant *Staphylococcus aureus* [34], and many others.

The differences between the membrane structures of Gram-negative and Gram-positive bacteria determine how easy and successful photoinactivation will be. In fact, Gram-negative bacteria have an outer membrane coated with lipopolysaccharides and an inner cytoplasmic membrane separated by a thin peptidoglycan layer, which is bound to the outer membrane by lipoproteins. This complex structure limits the drug penetration [35]. A further efficient defense mechanism enhancing antimicrobial resistance is the formation of a bacterial biofilm. Biofilms are structured 3D complexes of clustered bacteria adherent to a surface and embedded in an extracellular polymeric matrix composed by many polymer-based substrates, such as polysaccharides, proteins, amyloids, lipids, extracellular DNA, membrane vesicles, and humic-like substances. This structure hinders the penetration of antimicrobials as well as the action of the host immune system [36].

In summary, it is now well established that an efficient photo-antimicrobial system should possess the following characteristics [6]:-Positive charge for high-affinity binding to negatively charged bacterial cell membranes;-Low molecular weight or a structure that facilitates penetration through the biofilm matrix;-High ^1^O_2_ quantum yield;-High photostability;-No dark toxicity and/or mutagenicity towards host eukaryotic cells in the “therapeutic window” where microorganisms can be killed without damaging the surrounding cells.

### 2.3. Antimicrobial PTT Activity

As previously mentioned, another light-induced therapeutic treatment against bacterial infections is called antimicrobial photothermal therapy (aPTT) [37]. It relies on the heating of microbial pathogens at temperatures above 45 °C, at which the viability of most bacteria is impaired, which is achieved by irradiating a suitable photothermal agent. Photothermal antimicrobial agents are nanostructures with significant absorption in the visible and NIR regions that are able to transform the absorbed light into heat. They can be classified according to their chemical structure into plasmonic metals (such as Au, Cu, Pd, and Bi), carbon-based materials (such as carbon dots, GO, and graphene nanosheets), polymers (polydopamine, polyaniline, and polypyrrole), and semiconductors such as bP-NPs. The first target of the photothermal action has been reported to be the bacterial cell membrane: The photo-induced heat triggers the destruction of microbial membranes; the generation of ROS, which causes lipid peroxidation; and the denaturation of proteins, leading to the destruction of the pathogen [38].

An important side effect encountered during PTT is the heat-induced shock to the nearby healthy tissues that can be painful for patients. To minimize the use of high temperatures and optimize the antimicrobial effect, the combination of PTT with PDT has the potential to be a therapeutic procedure that is synergistic and highly effective.

Among emerging antimicrobial nanomaterials, bP nanosheets have a unique ability to absorb light across a broad spectrum ranging from UV to near-infrared light, producing singlet oxygen with a high quantum yield or heat with remarkable photothermal conversion efficiency [13]. These peculiar features make bP exploitable for PTT and PDT as well [16,39,40].

## 3. Structure and Properties of bP and bP-NPs

At standard temperature and pressure, bP is the most thermodynamically stable phosphorus allotrope. It was obtained for the first time by heating white phosphorus under high pressure (1.2 GPa). Compared to the other two allotropes—the amorphous red phosphorus and the notoriously unstable white phosphorus—bP has a higher density (2.69 g/cm^3^ vs. 2.05–2.34 g/cm^3^ and 1.83 g/cm^3^, respectively) and better thermal stability and can withstand temperatures up to 550 °C in the air without catching fire spontaneously [41].

bP has historically not attracted much interest from the scientific community until 2014, when it was shown that from its crystal structure, it is possible to obtain a new 2D material known as phosphorene, which is formed by a single layer of phosphorus atoms [42,43]. Phosphorene has an orthorhombic crystalline structure that contains eight atoms per unit cell. Similar to its precursor (bP), each layer is composed of P atoms held together by a strong covalent bond. However, along the stacking direction, the layers interact through weak van der Waals forces. A monolayer of bP is composed of P atoms with five valence electrons with 3s^2^3p^3^ configuration, each covalently bonded to the other three neighboring P atoms by their p-orbitals [44,45]. Each P atom is sp^3^-hybridized and bears a lone pair. The crystal structure of bP is shown in Figure 2.

Regarding the single layer, P atoms are not arranged in the same plane: Some are in the upper sublayer, and the others are in the lower sublayer, together constituting the unique puckered honeycomb crystal structure. The distance between the upper and lower sublayers is ~2.1 Å, and the distance between the two layers is about 5 Å [46]. bP belongs to the orthorhombic system with reduced symmetry, with the main crystallographic *b* axis normalized among the layers. In the layer plane, there are two characteristic crystal orientations, as shown in Figure 2: armchair (AC, parallel to the pucker) and zigzag (ZZ, perpendicular to the pucker), respectively [47,48,49,50].

This unique in-plane structural anisotropy leads to many intrinsic anisotropic electrical, optical, thermal, and mechanical properties that distinctly differentiate phosphorene from isotropic planar 2D materials [51] already performing potential applications in biomedicine.

### 3.1. Physical, Mechanical, and Chemical Features of bP and Phosphorene (bP-NPs)

bP is a semiconductor with a direct band gap, high carrier mobility, and thermal stability in vacuum at around 400 °C. It exhibits a tunable bandgap that varies with the number of layers, ranging from 0.3 eV for bulk to 2 eV for the monolayer. This range covers the spectrum between graphene and transition metal dichalcogenides, rendering bP-NPs an extremely appealing option as a 2D semiconductor [52,53,54]. Together with the bandgap, carrier mobility is layer-dependent, and it has been calculated to achieve the values of 10,000–26,000 cm^2^/Vs for the monolayer [55]. The well-known in-plane anisotropy generates peculiar physical and mechanical responses distinguishing between AC and ZZ directions.

For example, the prominent directions of electron transport and heat transport are orthogonal. This means that the electrical conductivity along the AC direction is greater than that along the ZZ direction, while the thermal conductivity is lesser in the AC direction than in the ZZ direction [55].

Anisotropy affects even the optical properties, showing the dichroic behavior of bP-NPs. In optics, dichroic materials absorb light differently based on its polarization. In the case of bP and bP-NPs, only light with a polarization component along the AC direction is absorbed for frequencies close to the bandgap energy [56]. Similar results have been recently experimentally obtained by measuring the photoluminescence behavior of bP-NPs [57].

bP exhibits unique mechanical properties due to its puckered structure. When stretched along the y-direction, it displays a negative Poisson’s ratio (−0.027) for the z-direction. This implies that when stretched along one direction, the material expands along the transverse direction, which is contrary to what happens with most materials, where stretching along one direction usually results in a reduction of lateral dimension [54,58]. Also, as bP is anisotropic, both theoretical and experimental studies have demonstrated different values for the Young’s modulus (E) and elongation at break along the ZZ and AC directions. For the monolayer, E_AC_ is about 25 GPa, and elongation at break is 30%, while along the ZZ direction, E_ZZ_ is around 100 GPa, and elongation at break is 27% [59].

Thanks to their electronic, thermoelectric, mechanical (Table 1), and optical properties, bP and bP-NPs have attracted the interest of scientists not just to deeply understand their properties but also to start developing and optimizing industrially scalable devices useful in electronics, photonics, engineering, and nanomedicine.

With reference to their actual and effective employment, it is necessary to point out that due to their peculiar structure, bP and especially bP-NPs are materials that exhibit high reactivity. This makes bP-NPs remarkably responsive to the surrounding environment and potentially suitable to be patterned and/or functionalized for specific performant applications, especially in the optoelectronics fields [46,48,53]. However, phosphorene reactivity under ambient conditions results in physical and structural changes, leading to degradation [60,61,62,63,64,65]. This behavior can be strictly related to the chemical structure of bP; indeed, each phosphorus atom is covalently bound to three other atoms, and each atom also has a dangling lone pair pointing out of the layer’s plane. The presence of these lone pairs causes bP to have a high affinity for oxygen, which can easily interact with the free electron doublets, ultimately leading to oxidative degradation of the layer [64].

The primary obstacle to the successful use of phosphorene in optoelectronics is the poor environmental stability despite intense recent research efforts aimed at its protection and passivation [59,61,63]. Such instability, as will be discussed later, advantageously becomes a beneficial feature for biomedical applications [62,64,66].

### 3.2. Preparation of bP and bP-NPs

Several synthetic procedures have been developed and tested for the preparation of phosphorene using both bottom-up and top-down approaches. Bottom-up methods directly synthesize bP-NPs from different molecular precursors through chemical reactions. Chemical vapor deposition, i.e., direct vapor deposition of red phosphorus or bulk bP in vacuum or argon, has not produced satisfactory results [50,67,68]. The precursors’ instability and procedural problems constitute a major limitation to the development of massive production of bP-NPs through bottom-up methods. Top-down methods involve separating the stacked layers of bulk bP to obtain single- or few-layered nanosheets by breaking the Van der Waals bonding. For this method, it is essential to begin with high-purity bulk bP, which can be synthesized using established methodologies. Bridgman first synthesized black phosphorus by heating white phosphorus at 200 °C under high pressure (1.2–1.3 GPa). In 2007, it was found that it could be prepared from red phosphorus at low pressure and 873 K by adding small quantities of gold, tin, and tin(IV) iodide [54]. All these processes can produce good-quality bP but are expensive and low-yielding. To date, the absence of a safe, high-throughput, and scalable route for producing bP remains one of the main limits to phosphorene uses. However, some companies, such as Smart-Elements GmbH (https://www.smart-elements.com/, accessed on 11 November 2023), are able to produce substantial quantities (in the range of hundreds of grams) at affordable prices, satisfying semi-industrial testing for some niche applications. The company’s website claims a production based on a modified vapor deposition method developed in its proprietary laboratories, which virtually eliminates the usual contaminants.

To produce bP-NPs, exfoliation is needed. Based on multilevel quantum chemical calculations, the exfoliation energy of bP is around 151 meV per atom (larger than that of graphite, 61 meV), which accounts for the relative difficulty in exfoliating bP. This is associated with a non-negligible electronic density overlap between the layers; indeed, it should be pointed out that there is debate regarding whether the interlayer bonding can be classified as a Van der Waals type. However, there are three main top-down methods:(1)Mechanical cleavage, also known as the “Scotch-tape” method, involves the sequential peeling off of layers from bulk bP using adhesive tape. After the process is complete, the material is transferred to a substrate (Si/SiO_2_) and cleaned. Although this technique can produce high-quality phosphorene, the yield is typically low, and contamination caused by adhesive residue cannot be ignored [43];(2)In electrochemical exfoliation, consisting of anodic oxidation and cationic intercalation, a voltage is applied to bulk bP, serving as an electrode in an electrolyte solution, causing a structural deformation of the layered bP and yielding 2D nanoflakes [50];(3)Sonication-assisted liquid phase exfoliation is a reliable method for producing high quantities of bP-NPs. This method consists of three steps: immersion in a solvent, ultrasonication, and purification [69].

When a solid, layered material is immersed in a liquid, the interfacial tension is significantly high so that the material–solvent interactions are not able to outweigh the interlayer interactions, and spontaneous exfoliation does not occur. It is indeed necessary to apply external energy to win secondary intra-layer interactions and exfoliate the material. Ultrasounds are used to generate microbubbles, the growth and collapse of which are attributed to the cavitation-induced pressure pulses and acoustic waves consisting of alternate regions of compression and rarefaction [70]. The choice of the solvent is crucial; it should have a surface tension similar to the surface energy of the 2D material to maximize the exfoliation rate and inhibit the restacking of nanosheets. In the case of bP, solvents with surface tensions of 35–40 mJ/m^2^ are used, such as dimethylformamide (DMF), dimethyl sulfoxide (DMSO), N-methyl-2-pyrrolidone (NMP), and N-cyclohexyl-2-pyrrolidone (CHP) [50,54]. Although anhydrous organic solvents may produce high-quality flakes, they have high boiling points, making postprocessing and disposal more difficult, and they are also hazardous to human health and the environment. After the exfoliation process, if successful, solvent molecules surround the nanoparticles through solvation, which stabilizes them. These molecules are challenging to remove, and they remain even after subsequent centrifugation and redispersion steps that are essential for a final collection of the products. Therefore, particularly in biomedical fields, this can cause safety issues. For this reason, water-based solutions have also been studied as possible sonication mediums. Since water has a surface tension of about 73 mJ/m^2^, and bulk bP is insoluble in water, stabilizing surfactants are needed to produce stable flakes and avoid aggregation. It is also important to use de-oxygenated water to prevent phosphorene from oxidizing. Some of the studied surfactants are polyvinyl alcohol [71], polyvinyl pyrrolidone [72], and sodium dodecyl sulfate. Moreover, the significant issues associated with light sensitivity [14] affect bP storage and pose stability challenges for practical use along with additional concerns of spontaneous oxidation and aggregation.

## 4. Antimicrobial Photoactivity of bP

After its discovery, bP attracted the interest of researchers mainly due to its possible applications in optoelectronics, photonics, and advanced engineering, and only in the last few years has it also emerged as a possible new 2D material for biomedical applications. bP is highly biodegradable, biocompatible, and safe for use. These properties are essential for the use of bP in medicine [46,48,73,74,75,76,77,78,79]. In living organisms, phosphorus is a crucial element that constitutes approximately 1% of the total body weight. When it degrades, it transforms into harmless phosphate, which exhibits high biocompatibility and low cytotoxicity, preventing its in vivo accumulation. As a 2D material, it intrinsically has a large surface area, making it suitable for the absorption of drug molecules and making it easier to control the kinetics of release [80]. It also has a high modulus; thus, it can be used to improve the mechanical strength of biomedical implants.

However, even if bP has been studied and proposed for many biomedical applications, such as bioimaging, biosensing, and theranostics, maybe one of the most interesting properties of bP-NPs is the rapid response to external signals such as light and heat, making it suitable for applications in phototherapy, namely for cancer treatment [81,82]. Recently, studies on its potential antimicrobial applications, especially against antibiotic-resistant pathogens, have been gaining growing attention.

### 4.1. Mechanisms of bP Photoactivity

As previously mentioned, bP has a bandgap dependent on the number of layers, which varies from 0.3 eV to 2 eV, from bulk to monolayer. The ^3^O_2_/^1^O_2_ redox potentials fall within this range [64], and thus, bP-NPs can act as light absorbers, mediating the energy transfer to oxygen molecules in the surrounding environment. Molecular dynamic simulations help to predict the mechanism of singlet oxygen (^1^O_2_) generation: First, oxygen molecules interact with P lone pairs, leading to the adsorption of O_2_, and then, the generation of ^1^O_2_ occurs through charge transfer [14]. The so-generated ^1^O_2_ is very unstable and reacts with target systems in the surroundings. Illumination is needed to excite the transition from the ground state of bP-NPs. The most convenient wavelength range for the photosensitizer activation is between 600 and 800 nm, which is called the “optimal therapeutic window”, since it is therapeutically safe and allows for effective tissue penetration of light while still providing enough energy to allow the transition to the excited singlet state of oxygen [23] without compromising biological tissues. Furthermore, bP can act as an excellent NIR light-responsive photothermal antibacterial agent owing to its strong NIR light-absorption properties and to its high photothermal conversion efficiency [83]. In this case, the bP-NPs are generally irradiated by NIR light between 800 and 1000 nm.

### 4.2. Bare bP-NPs

The antimicrobial activities of exfoliated bP nanosheets have been compared to those of bulk bP and other 2D materials such as graphene and transition metal dichalcogenides such as MoS_2_, showing a better performance in killing both Gram-negative *Escherichia coli* and Gram-positive *Staphylococcus aureus* that cause serious infections. Under 808 nm laser irradiation, a value of 99.2% in bacterial killing percentage against both *E. coli* and *S. aureus* was reached mainly by means of photothermal inactivation with a negligible cytotoxicity towards mammal cells even at high bP-NP concentrations [84]. A few nanograms of bP nanosheets have been shown to be enough for a strong and broad-spectrum antimicrobial activity toward the bacteria *Escherichia coli*, *Pseudomonas aeruginosa*, MRSA, *Salmonella typhimurium*, and *Bacillus cereus* as well as the fungal strains *Candida albicans*, *Candida auris*, and *Cryptococcus neoformans*, displaying the effectiveness of bP-NPs as an antibacterial additive in surface coatings, too. The main mechanism of antimicrobial activity is the production of ROS species, superoxide radicals, and ^1^O_2_, which are able to cause cell oxidation with membrane disruption and the resulting cell lysis [85].

High-resolution microscopy and ATR-FTIR studies have revealed that the physical interaction of the bP-NPs with the microbial membranes, together with the oxidative stress, cause important physical and biochemical damages to the phospholipids and to the amide I and II proteins, whereas this results in slight chemical modifications to polysaccharides and nucleic acids of Gram-positive methicillin-resistant *Staphylococcus aureus* (MRSA) and Gram-negative *Pseudomonas aeruginosa* and to the fungal species *Candida albicans* [86]. Recently, bP-NPs have been tested against in vitro cultures of *S. aureus* and *P. aeruginosa* as well as an in vivo preclinical model of an acute murine wound infection of *S. aureus*. For the first time, it was demonstrated that bP-NPs can have significant in vivo antimicrobial effects through oxidation in ambient light conditions without the need for NIR light irradiation (Figure 3). The treatment did not produce any secondary effects, such as tissue inflammation, toxicity, or necrosis in the mice organs. These results open new scenarios in clinical wound management [87].

### 4.3. bP-NPs-Based Hybrid Materials

bP has also emerged as a suitable nanomaterial that allows for drug delivery and therapeutics due to its high surface area. As it is negatively charged in water with an interlay distance of ~5.24 Å, the encapsulation of small and positively charged molecular drugs within the interlayer spaces is possible mainly through electrostatic interactions. To date, three major strategies have been explored for developing bP-NPs-based hybrid materials: electrostatic interaction, covalent bonding, and noncovalent bonding (e.g., hydrophobic interactions) [83].

Due to the synergistic action of several photoactive and/or antimicrobial materials, bP-based hybrid nanocomplexes show better antibacterial activity than the lone bP-NPs.

#### 4.3.1. Metals

Silver nanoparticle (AgNP)-doped bP-NPs were developed as an economic and potent synergistic disinfectant activated by solar radiation for a rapid disinfection/wound healing, and they have also shown a great potential for environmental remediation applications through photo-induced disinfection of water and degradation of pollutants. It has been hypothesized that the AgNPs facilitate the adsorption and activation of O_2_, thus enhancing the photogeneration of ROS on bP-NPs–AgNP nanohybrids and strongly increasing the affinity toward bacteria, leading to a synergistic pathogen inactivation. On the other hand, the complexation with bP-NPs prevents AgNPs from self-aggregation when in contact with bacterial membranes and limits excessive consumption of Ag, which may result in gastrointestinal disorders, spasms, and in some cases also death [88].

bP-NPs modified with titanium aminobenzenesulfonato (Ti-SA4) (an antibacterial metal complex) have been shown to improve the resistance of bP-NPs to oxidation and to aid in the interaction with negatively charged bacterial membranes, increasing the therapeutic effect against the two common pathogens Gram-negative *Escherichia coli* and Gram-positive *Staphylococcus aureus* [89].

The antibacterial and antibiofilm photoactivities of bP–gold nanocomposites (bP–Au NPs) were investigated against *Enterococcus faecalis*, an opportunistic pathogen that is the main cause of nosocomial infections such as endocarditis, urinary tract infection, bacteremia, peritonitis, and prosthetic joint infection. Under NIR light irradiation at 808 nm, the photothermal effect and oxidative stress caused bacterial eradication by destroying the microbial cell membrane and inhibiting biofilm formation by up to 58% [90].

Combining the antibacterial features of AuNPs and ZnO with the photoactive properties of bP-NPs, a new NIR light-responsive synergistic nanoplatform, Au–ZnO–bP, was prepared and its antibacterial activity against *S. aureus* species assessed under NIR irradiation. It is worth mentioning that the powerful effect of this bP–hybrid nanosystem on bacterial resistance suppression was observed even after 10 consecutive treatments [91].

#### 4.3.2. Hydrogels

Hydrogels are hydrophilic polymers with three-dimensional networks widely used as wound dressing. In fact, they offer a humid microenvironment that allows a faster healing of skin wounds, they absorb tissue exudates, and they protect wounds from infection. A novel strategy for rapid wound healing was explored by conjugating bP-NPs with chitosan (CS) hydrogel through electrostatic interaction for application in aPDT. The bP-NPs in this hydrogel system were shown to rapidly generate ^1^O_2_ under visible light, improving the antibacterial efficacy against both Gram-positive and Gram-negative bacteria. Furthermore, the designed hydrogel system was found to be able to simultaneously stimulate the formation of fibrinogen and the cellular proliferation and differentiation needed to promote skin regeneration and wound healing [92]. Diabetic wound regeneration has been successfully promoted by PTT and PDT through utilizing a derivative of the metabolite itaconate 4-octyl itaconate (4OI), which has antioxidant and anti-inflammatory properties, complexed with bP-NPs and incorporated into a photosensitive, multifunctional gelatin methacrylamide hydrogel (4OI-BP@Gel). bP-NPs acting as a carrier were capable of releasing 4OI in proximity to endothelial cells, promoting angiogenesis and facilitating skin-injury healing (Figure 4) [93].

A similar therapeutic approach was performed utilizing epigallocatechin gallate (EGCG)-modified bP quantum dots (BPQDs) incorporated into hydrogel under NIR light. In fact, green tea polyphenols such as EGCG have antioxidant, antibiosis, anticancer, radiation resistance, and immunity enhancement properties that improve wound healing [94].

Recently, a thermosensitive bP hydrogel prepared using chitosan, β-glycerophosphate disodium salt, and hydroxypropyl cellulose was loaded with silver sulfadiazine, an antibacterial drug used for burn-wound treatment, promoted skin-wound healing. Thanks to a synergistic phototherapeutic and antibacterial treatment, under NIR irradiation, silver sulfadiazine was continuously released, promoting collagen deposition, enhancing neovascularization, and reducing inflammatory markers (Figure 5) [95].

#### 4.3.3. Antimicrobial Compounds

A light-responsive antibacterial nanoplatform was developed by incorporating the pharmaceutical antimicrobial vancomycin and BPQDs into a thermal-sensitive liposome for synergistic photothermal and pharmacological therapy of skin infection under NIR light. Due to the photothermal effect of BPQDs, the heat-sensitive liposome is broken following NIR irradiation, and the antibiotic vancomycin is directly released into the infected skin abscess, killing bacterial pathogens such as methicillin-resistant *Staphylococcus aureus* (MRSA). Therefore, photo-induced heat generated by BPQDs could effectively eradicate bacteria (Figure 6) [96].

bP-NPs loaded with the antimicrobial physcion (Phy), a naturally occurring anthraquinone derivative, through hydrophobic interactions have been shown to achieve a good stability and low cytotoxicity under physiological conditions. After irradiation at 808 nm, the photo-induced heat from bP-NPs caused the release of the physcion so that, thanks to the synergistic photothermal and antimicrobial effects, 99.7% of *S. aureus* as well as 99.9% of *P. aeruginosa* were killed [97].

Antimicrobial peptides are an emerging class of pharmaceuticals. Among them, the naturally cationic polyamide *ϵ*-poly-l-lysine (*ϵ*-PL), consisting of 25–30 lysine residues, has been extensively employed as a food preservative and against Gram-positive and Gram-negative bacteria species, yeasts, and molds. Due to its large positive charge, *ϵ*-PL can strongly interact with and destroy the negatively charged bacterial membranes. Through electrostatic interactions, *ϵ*-PL has been complexed with bP-NPs in order to achieve dual photo- and antimicrobial activity against multidrug-resistant bacteria [98].

A novel approach to fighting infective diseases is to focus particular attention on intracellular pathogens. Almost all bacteria are able to hide and survive inside blood-derived phagocytes, causing the infection to be either latent or recurrent. Silent-infected macrophages can be a “Trojan horse” and in certain circumstances may elicit pulmonary infections and endocarditis. Consequently, drugs that have the ability to target macrophages are particularly crucial for a complete bacterial eradication. For this purpose, mannosylated bP nanosheets (Man-bP NPs) were used as a drug carrier and loaded with the photosensitizer chlorin e6 (Ce6) to obtain the complex Ce6@Man–BPN. The potential loading mechanism mainly lies in the hydrophobic interaction between them. It was shown to preferably bind the mannose receptors on the surface of infected macrophages and release bP-NPs and Ce6 inside the cells; both the nanoparticles and drug can be photoactivated for killing bacteria by means of synergic PTT and PDT, with negligible toxicity to mammalian cells (Figure 7) [99].

## 5. Conclusions and Future Perspectives

In this review, we explored the structural and physicochemical properties as well as the potential and versatility of the use of bP-NPs both alone and conjugated with other drugs, photosensitizers, or metals in antimicrobial phototherapy and attempted to provide an overview of the variety of studies carried out so far.

The advancements in biomedical research and the development of nanotechnology can have great potential for fighting multi-drug-resistant microbial pathogens. Among 2D nanomaterials, bP-NP has shown promising properties for various applications, including antimicrobial phototherapy. It has a high specific surface area, which can enhance its interactions with microbial pathogens. The puckered structure of bP-NPs along with their peculiar anisotropy contribute to unique electronic, mechanical, and photoactivity properties that are potentially exploitable in an antimicrobial system. The thickness-dependent band gap (0.3–2 eV) allows it to absorb a wide range of wavelengths from ultraviolet to NIR. Due to its broad absorption band and unique electronic structure, bP-NP possesses an intrinsic photoactivity that makes it an effective phototherapeutic agent in PTT and PDT against pathogenic bacteria. Furthermore, bP nanomaterials have excellent biocompatibility and biodegradability in vivo, minimizing potential adverse effects on living tissues and facilitating their use in antibacterial activity as well as other biomedical applications. Phosphorene-based nanomaterials can be engineered for targeted drug delivery. This is particularly relevant in antimicrobial phototherapy, where the precise delivery of therapeutic agents to the infection site is crucial for maximizing treatment efficacy and minimizing side effects.

Along with their many discussed advantages in antimicrobial applications, there are still several challenges facing bP-NPs that need to be addressed, such as the need for efficient and low-cost synthesis strategies to facilitate large-scale production, stability optimization for ensuring efficacy in biomedical applications, and systematic studies correlating structural characteristics with the antimicrobial properties. In fact, although numerous current studies and insights on the structure, chemical, and physical characteristics of bare bP-NPs are present, the research on applications of bP-NPs-based hybrids in aPDT and aPTT often lacks clear reports on the structure/chemical composition of the investigated hybrid systems and on the mechanisms involved in the photoactivation phenomena. The modification and embedding of bP-NPs in different substrates, such as other photosensitizers, drugs, and protective agents, to create hybrid systems can complicate the identification and rationalization of individual components’ contributions to photoactivation processes. In addition, PDT and PTT are often used in combination with each other, simultaneously causing ROS formation and temperature rise. Therefore, finding the correct structure/property correlation that allows for a detailed analysis of the photophysical phenomena that create the antimicrobial effects remains challenging.

The future perspectives for phosphorene in antimicrobial phototherapy are promising, and ongoing research suggests that phosphorene-based materials hold potential for further advancements in the field. Continued efforts to explore scalable and cost-effective production techniques are crucial for making phosphorene more accessible for medical applications. Investigating and developing hybrid materials that combine phosphorene with other nanomaterials, drugs, or photosensitizers can lead to synergistic effects, providing a comprehensive approach to combatting microbial infections. Fine-tuning certain properties of phosphorene, such as its band gap, surface functionalization, and size, may improve the targeted delivery and enhance therapeutic effects. A better understanding of the pharmacokinetics, biodistribution, and long-term biocompatibility of phosphorene-based materials could pave the way for clinical trials and eventual translation into practical medical treatments. In addition, investigating the synergistic effects of phosphorene nanomaterials by combining photothermal and photodynamic approaches could lead to more effective and versatile APT strategies.

## Figures and Tables

**Figure 1 pharmaceutics-15-02748-f001:**
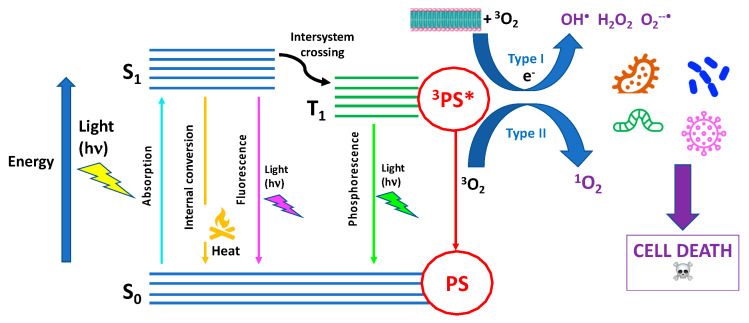
Scheme of Jablonski diagram illustrating the fundamental photochemical and photophysical mechanisms of antimicrobial photodynamic therapy (aPDT).

**Figure 2 pharmaceutics-15-02748-f002:**
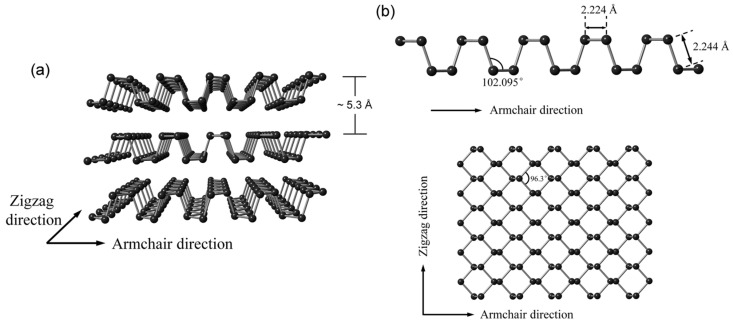
Crystal structure of phosphorene with the two characteristic crystal orientations (**a**) zigzag (ZZ) and armchair (AC) and (**b**) corresponding angles and bond lengths. Figure reproduced from [44] with permission from the Royal Society of Chemistry.

**Figure 3 pharmaceutics-15-02748-f003:**
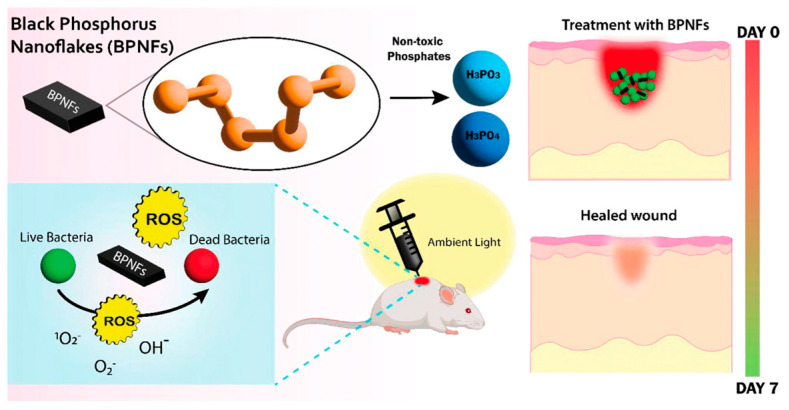
Scheme of the effects of bP-NPs on wound healing upon exposure to ambient light. Figure reprinted from [87] under Creative Commons CC BY license.

**Figure 4 pharmaceutics-15-02748-f004:**
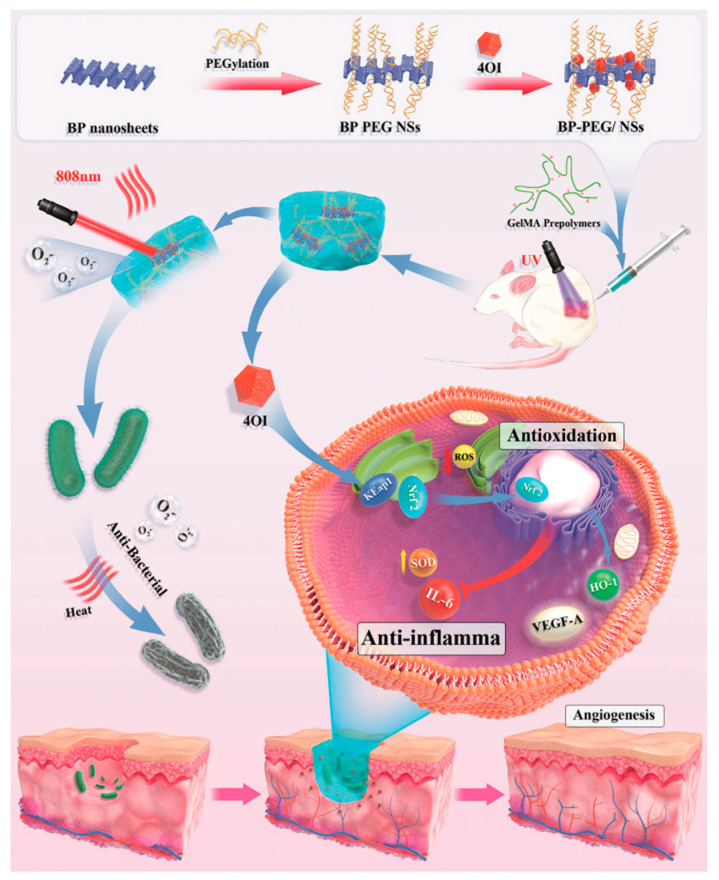
Schematic illustration of the effects of the photosensitive, multifunctional gelatin methacrylamide hydrogel (4OI-BP@Gel) on wound healing. Reprinted with permission from [92] Copyright 2022, John Wiley and Sons, Hoboken, NJ, USA.

**Figure 5 pharmaceutics-15-02748-f005:**
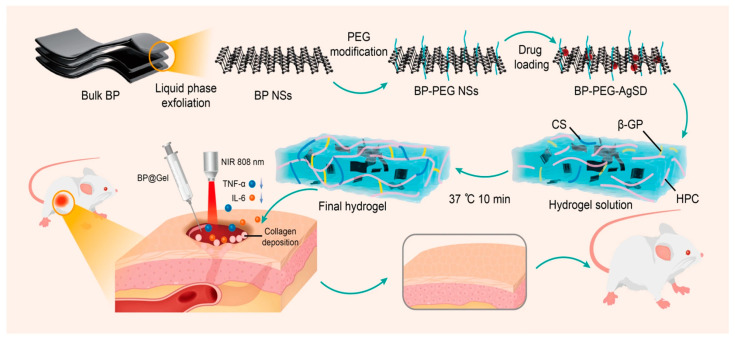
Schematic illustration of the mechanism of thermosensitive bP hydrogel prepared using chitosan (CS), β-glycerophosphate disodium salt (β-GP), and hydroxypropyl cellulose (HPC) loaded with silver sulfadiazine (AgSD) for skin-wound healing. Reprinted from [94] under a Creative Commons Attribution 4.0 International License.

**Figure 6 pharmaceutics-15-02748-f006:**
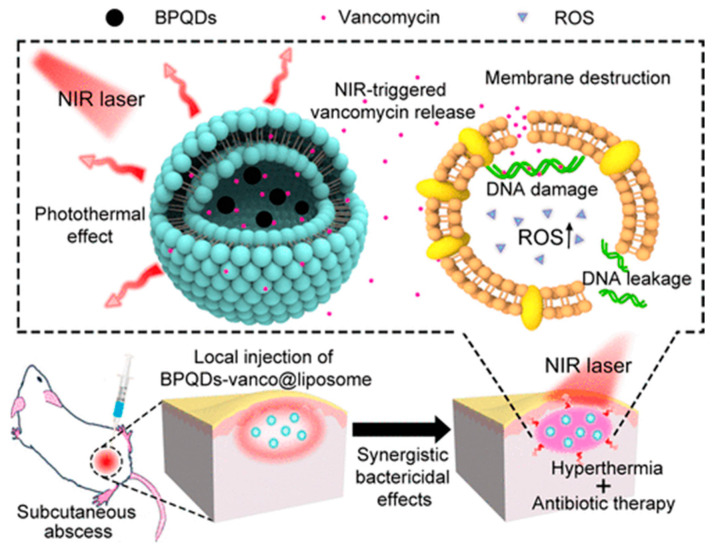
Schematic Illustration of the photon-responsive antibacterial nanoplatform for synergistic treatment of a subcutaneous abscess. Reprinted with permission from [96] Copyright © 2018 American Chemical Society.

**Figure 7 pharmaceutics-15-02748-f007:**
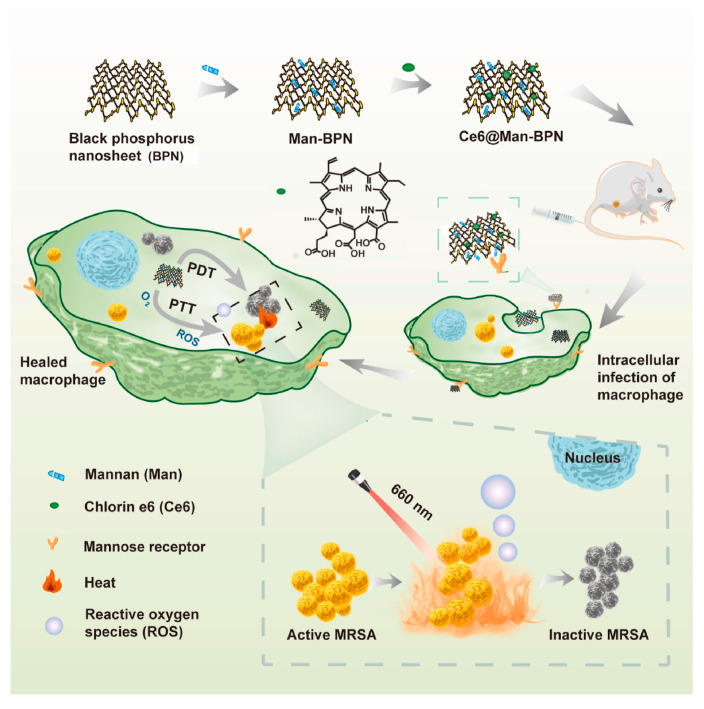
Schematic Illustration of the mechanism of action of the mannosylated bP nanosheet loaded with the photosensitizer chlorin e6, i.e., Ce6@Man–BPN, in targeting macrophages and inactivating intracellular bacteria. Reprinted from [99], Copyright 2023, with permission from Elsevier, Amsterdam, The Netherlands.

**Table 1 pharmaceutics-15-02748-t001:** Properties of bP and bP-NPs.

Properties	bP ÷ bP-NPs	
Band gap (eV)	0.3 ÷ 2	
Band type	Direct	
Charge mobility (cm^2^/Vs)	1000 ÷ 600	
Seebeck coefficients (μV/K at 300 K)	413/300 ÷ 50	
Thermal conductivity (W/mK)	**ZZ**	**AC**
	~18	12
Elongation at break (%)	27	30
Young’s modulus (GPa)	~100	~25

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
