# Peer review of "Innovative Phosphorene Nanoplatform for Light Antimicrobial Therapy"

_pharmaceutics, 2023, doi:10.3390/pharmaceutics15122748_

Round 1
Reviewer 1 Report
Comments and Suggestions for Authors
In this review, the authors explored the structural and physicochemical properties of nanoparticles and their potential versatility in antimicrobial phototherapy. They discussed the promising properties of NPs, such as high specific surface area, a unique electronic structure, and biocompatibility, making them effective agents in photothermal and photodynamic therapies against pathogenic bacteria. The review highlighted challenges, including the need for efficient synthesis strategies, stability optimization, and a lack of clear reports on the structure/chemical composition of hybrid systems. Despite these challenges, the authors emphasized the promising future perspectives in antimicrobial phototherapy, with ongoing research suggesting potential advancements. They underscored the importance of scalable and cost-effective production techniques, exploring hybrid materials, and fine-tuning phosphorene properties for targeted delivery. The authors called for a better understanding of pharmacokinetics and biodistribution, paving the way for clinical trials and practical medical treatments. They concluded by noting the potential of combining photothermal and photodynamic approaches for more effective and versatile antimicrobial phototherapy strategies. Apart from this, the contents of the manuscript are described properly, and necessary illustrations are included. This can serve as a good guide for researchers in this field, and in my opinion, it can be accepted for publication. Some editorial corrections are required, otherwise, it's fine to be accepted.
Author Response
We thank very much the reviewer for reading carefully, understanding our intentions, and appreciating our review.
Reviewer 2 Report
Comments and Suggestions for Authors
The manuscript entitled “Inovative phosphorenes nanoplatforms for light antimicrobial therapy» is submitted by A. Sgarbossa for publication in Pharmaceutics. In this review, the photophysical properties of phosphorene (“bp-NP”, a 2D-nanomaterial obtained by exfoliation of black phosphorus, bP) are described in the perspective of use alone (with band gap and related absorption properties depending on the number of layers) OR as nanoplatform for conjugation with photo-sensitizers (PSs), in antimicrobial photodynamic therapy (aPDT) and photo-thermal therapy (PTT). Potential synergistic effects are also highlighted in this review.
Remarks and comments:
- Figure 1, please add an arrow on the left for increasing energy.
- Please check the literature again; may be references are missing ; for example Pharmaceutics, 2021, 13, 1344 (also related to the use of bP as nano-platform in medicine)
More generally and as a conclusion, this manuscript is well written and quite complete; the schemes are clean; this study (report) corresponds to a review in a field of interest, and I personally think it could deserve for publication in Pharmaceutics; may be more precise information could be given related to the nature of the PSs - bp-NPs “conjugation”; is there any covalent bonds, on other interactions (VdW, ionic ….); please, culd the authors give more details?
Author Response
We thank the reviewer for the time assessing our manuscript and helping us to improve it.
We have added an arrow on the left of Figure 1 as requested and we have inserted the reference “Pharmaceutics, 2021, 13, 1344” [80].
Concerning the information on the nature of the interaction between drugs/ photosensitizers and bP-NPs we have added a general sentence at lines 420-427 and we have specified that in the case of complex Ce6@Man-BPN, Chlorin 6 interacts with bP mainly by means of hydrophobic forces (Iine 533). As stated in the conclusions, unfortunately the current studies on bP-NPs-based hybrids often lack clear reports on the structural/chemical characteristics.
Reviewer 3 Report
Comments and Suggestions for Authors
In this review the authors report the physicochemical properties and the possible the use of phosphorene-based nanomaterials, alone and conjugated with other drugs in the antimicrobial phototherapy.
bP-NP is a 2D nanomaterials which due to it high specific surface area, can enhance interaction with microbial pathogens, moreover it have excellent biocompatibility and biodegradability in vivo. Among the disadvantages known for antimicrobial applications of this materials there is the a high cost and not very efficient synthesis.
The review is interesting but appears too unbalanced in the introductory part which is excessive. It concerns aspects relating to the production and the structural and chemical-physical characteristics of the material with a wealth of detail that is probably excessive compared to the part concerning pharmacological applications. The authors could/should balance the two parts better. Then a careful rereading is necessary to eliminate some typographical errors.
Comments on the Quality of English LanguageA careful rereading is necessary to eliminate some typographical errors.
Author Response
We thank very much the reviewer for the appreciation of our work.
We agree that the part on the production and the structural and chemical-physical characteristics of bP-NPs probably contains too many details. So, we have eliminate excessive information (lines 226-227; 232; 236; 240-241; 245; 248; 263; 287-288; 328). We have carefully re-read the paper as suggested.
Reviewer 4 Report
Comments and Suggestions for Authors
1. The introduction needs to be divided into paragraphs. e.g. by breaking the text at lines 32, 44, 59, 64, 69
2. Lines 81 and 188 and 572. The absorption of light is not even across the range of wavelengths, and this important point needs to be shown. Please create and include an image showing the absorption spectrum of bP from UV to visible to near infrared, e.g. 200-1000 nm. Ref 13 is cited in line 81 but this only discusses 300-600 nm. Ref 39 is better as this shows data for 200-1000 nm. Actually, bP has its strongest absorption from 200-400 nm, so the text on line 81 should be discussing UV light, which is overlooked.
2. Lines 231 and 282. There are no optical properties shown in table 1 so delete the word optical from line 282.
3. Line 297. Ref 13 shows the incredible problems of sensitivity to light which affect storage in various forms. This does pose an issue for stability of these materials for practical use. Adding to this are problems of spontaneous oxidation and aggregation. Line 373 should have a comment to recognise these practical issues.
4. Line 404. The text states "The proper wavelength range for the photosensitizer activation is between 600-800 nm". This is not the wavelength range for most effective aPDT activation but rather the convenient therapeutic window for tissue penetration of light. Hence the text needs to make that point clear.
5. Line 420. Under 808 nm irradiation, the main mechanisms will be photothermal (PTT) not photodynamic (aPDT). (This is correctly stated in line 470).
6. Line 252. The correct word is disinfection, not sterilization.
7. Line 459. The correct word is consumption or ingestion, not assumption.
8. The references have style inconsistencies (journal names and use of capitals in article titles).
Author Response
We thank very much the reviewer for the appreciation of our work. We have made all the corrections suggested
- The introduction needs to be divided into paragraphs. e.g. by breaking the text at lines 32, 44, 59, 64, 69.
We divided into paragraphs
- Lines 81 and 188 and 572. The absorption of light is not even across the range of wavelengths, and this important point needs to be shown. Please create and include an image showing the absorption spectrum of bP from UV to visible to near infrared, e.g. 200-1000 nm. Ref 13 is cited in line 81 but this only discusses 300-600 nm. Ref 39 is better as this shows data for 200-1000 nm. Actually, bP has its strongest absorption from 200-400 nm, so the text on line 81 should be discussing UV light, which is overlooked.
We have specified better this point substituting visible with UV (lines 84 and 191), we have put the reference 39 as suggested and now it is [13] (lines 85 and 193). We don't think it's necessary to add another figure as the range of absorption has already been stated correctly.
- Lines 231 and 282. There are no optical properties shown in table 1 so delete the word optical from line 282.
We did it.
- Line 297. Ref 13 shows the incredible problems of sensitivity to light which affect storage in various forms. This does pose an issue for stability of these materials for practical use. Adding to this are problems of spontaneous oxidation and aggregation. Line 373 should have a comment to recognise these practical issues.
We have written a sentence about this issue (lines 345-347)
- Line 404. The text states "The proper wavelength range for the photosensitizer activation is between 600-800 nm". This is not the wavelength range for most effective aPDT activation but rather the convenient therapeutic window for tissue penetration of light. Hence the text needs to make that point clear.
We have clarified this important point (lines 378-380)
- Line 420. Under 808 nm irradiation, the main mechanisms will be photothermal (PTT) not photodynamic (aPDT). (This is correctly stated in line 470).
We have corrected this point (393-401).
- Line 252. The correct word is disinfection, not sterilization.
We did it
- Line 459. The correct word is consumption or ingestion, not assumption.
We have corrected this word.
- The references have style inconsistencies (journal names and use of capitals in article titles).
We have checked automatically the references according to the style requested by the journal.
Round 2
Reviewer 3 Report
Comments and Suggestions for Authors
The manuscript has been carefully revised and improved in the parts that appeared redundant, in the opinion of this reviewer it can be accepted for publication in Pharmaceutics.